

# Prevalence and virulence genes of *Staphylococcus aureus* from food contact surfaces in Thai restaurants

Kannipa Tasanapak[1,2], Siriwat Kucharoenphaibul[1],
Jintana Wongwigkarn[1], Sutthirat Sitthisak[1], Rapee Thummeepak[1],
Plykaeow Chaibenjawong[1], Wassana Chatdumrong[1] and
Kunsuda Nimanussornkul[3]

[1] Microbiology and Parasitology, Naresuan University, Muang, Phitsanulok, Thailand
[2] Centre of Excellence in Medical Biotechnology, Faculty of Medical Science, Naresuan University, Muang, Phitsanulok, Thailand
[3] Faculty of Economics, Chiang Mai University, Muang, Chiang Mai, Thailand

## ABSTRACT

**Background:** *Staphylococcus aureus* is one of the most common pathogens responsible for food poisoning due to its ability to produce staphylococcal enterotoxin (SE). *S. aureus* can form biofilms on the surfaces of food processing devices, enabling the distribution of SE on foods through cross-contamination events. Thailand is known for its exotic cuisine, but there is no data on the prevalence of SE-harboring *S. aureus* in restaurants in Thailand.
**Methods:** In this study, we conducted surface swabs on surfaces of kitchen utensil that come into contact with food and on the hands of food handlers working in restaurants in the north part of Thailand. Isolated *S. aureus* was investigated for biofilm formation, virulence, and SE genes.
**Results:** Two hundred *S. aureus* were isolated from 650 samples. The highest prevalence of *S. aureus* contamination was detected on the hands of food handlers (78%), followed by chopping boards (26%), plates (23%), knives (16%), spoons (13%), and glasses (5%). All of them were methicillin-sensitive *S. aureus* (MSSA) and the *mecA* gene was not present in any strains. Biofilm formation was detected using the CRA method, and 49 (24.5%) were identified as biofilm-producing strains, with the hands of food handlers identified as the primary source of biofilm-producing strains. The prevelence of biofilm-related adhesion genes detected were: *ica*AD (13%), *fnb*A (14.5%), *cna* (6.5%), and *bap* (0.5%). Two classical enterotoxin genes, *sec* and *sed*, were also found in four and six of the *S. aureus* isolates, respectively, from hands and utensils.
**Conclusion:** The highest prevelence of *S. aureus* was detected on the hands of food handlers. *S. aureus* strains with biofilm and enterotoxin production abilities were discovered on food contact surfaces and the hands of food handlers, implying significant risk of food contamination from these sources that could be harmful to consumers. To avoid cross-contamination of food with food contact items, the food handlers' hands should be properly washed, and all food preparation equipment should be thoroughly cleaned.

Corresponding author
Kannipa Tasanapak,
kannipal@nu.ac.th

# INTRODUCTION

*Staphylococcus aureus* (*S. aureus*) is a significant pathogen generally found on human bodies, medical equipment, and contact surfaces in both hospital and community environments. It is the predominant species in surgical site infections (*Al-Awaysheh, 2018*), and is also found in food contact containers and food handlers in hotels and restaurants (*Beyene et al., 2019*; *Plaza-Rodriguez, Kaesbohrer & Tenhagen, 2019*; *Shahid et al., 2021*). It was found that only one CFU or low amount of *S. aureus* came from transfer of raw chicken meat to food-contact equipment, or from cross-contamination and recontamination to serving utensils (*Plaza-Rodriguez, Kaesbohrer & Tenhagen, 2019*). The pathogenesis of *S. aureus* strains is due to a combination of virulence factors such as extracellular factors, toxins, adhesion, and biofilm formation (*Khoramian et al., 2015*; *Cheung, Bae & Otto, 2021*). Biofilm formation helps protect the pathogen from phagocytes, while synthesis of toxins eliminates neutrophils and other leukocytes (*Cheung, Bae & Otto, 2021*). *S. aureus* strains have been found to be increasingly resistant to antibiotics, especially methicillin, making them one of the most difficult-to-treat pathogens (*Phokhaphan et al., 2017*; *Hamdan et al., 2022*). *S. aureus* can cause food poisoning, with instances of staphylococcal food poisoning resulting from the consumption of food contaminated with *S. aureus* carrying staphylococcal enterotoxin (SEs) reported in several countries (*Kavinum, Peanumlom & Wongchai, 2017*; *Watcharakorn, 2018*; *Sato'o et al., 2014*). In Thailand, food poisoning outbreaks caused by eating food contaminated with *S. aureus* occur frequently (*Kavinum, Peanumlom & Wongchai, 2017*; *Watcharakorn, 2018*). Although various enterotoxins have been reported, staphylococcal enterotoxin A (SE-A) and staphylococcal enterotoxin D (SE-D) are the most important because of their heat resistance properties (*Hwang et al., 2007*; *Pelisser et al., 2009*). SE-A's ability to retain some biological activity after heating to 121 °C indicates that the heating process of routine cooking is not sufficient to destroy them (*Balaban & Rasooly, 2000*). Most SE genes are located on mobile genetic elements, thus encouraging distribution of SE genes to other strains (*Pinchuk, Beswick & Reyes, 2010*). When *S. aureus* appears on utensils or the hands of food handlers, the chances of it contaminating food and causing disease are unavoidable (*Ciccio et al., 2015*; *Castro et al., 2016*).

   *S. aureus* can produce biofilm on food processing surfaces such as polystyrene and stainless steel and on human hands (*Ciccio et al., 2015*; *Ballah et al., 2022*). Biofilms are a problem in the food industry as they make sufficient cleaning and disinfection of surfaces contaminated with bacteria and organic matter difficult (*Avila-Novoa et al., 2018*). In Thailand, the survival of methicillin-resistant coagulase-negative staphylococci (MR-CoNS) in harsh conditions, such as on the surface of animated objects and dust in university and hospital environments, has also been reported (*Kitti et al., 2019*; *Seng et al., 2017*; *Chaibenjawong et al., 2022*). Many of these MR-CoNS strains were found to carry genes linked to biofilm formation (*Kitti et al., 2019*; *Seng et al., 2017*). The presence of

biofilm-related genes in *S. aureus* allows intramammary adherence and biofilm formation, leading to the inefficacy of antibiotic treatment and to chronicity (*Cucarella et al., 2001*).

This study aims to detect the prevalence of *S. aureus* contamination from food handlers and food contact surfaces (plates, glasses, spoons, chopping boards, and knives), as well as detect the rate of biofilm formation and the presence of biofilm-related genes (*e.g.*, *ica*AD, *fnb*A, *cna*, and *bap*) and classical enterotoxin genes (*sea*, *seb*, *sec*, *sed*, and *see*). The results of this study can help improve awareness of *S. aureus* contamination and help in its prevention, allowing restaurants to pay more attention to food hygiene and consumer safety.

## MATERIALS AND METHODS

### Samples collection

This study was approved by the Institutional Review Board of Naresuan University (IRB No. P10203/64). Written consent forms were signed by all participants. Prior to collecting samples, all serving utensils and food handler hands were cleaned using the methods commonly used in their restaurants, with surfaces being allowed to dry before taking swabs. The swab method of sampling was then carried out, as described in Microbiological Quality Criteria for Food and Food Contact Containers issued by the Department of Medical Science, Ministry of Public Health, Thailand (2017). Each testing swab stick was removed from its sterile wrappings and the tip was moistened by immersing it in 0.1% peptone water. After wiping the surfaces of plates, spoons, chopping boards, hands, knives, and glasses with the swab sticks, the sticks were dipped back into 0.1% peptone water. Altogether, 650 swab samples were collected from hands (150 samples), plates (100 samples), spoons (100 samples), knives (100 samples), chopping boards (100 samples), and glasses (100 samples) in local restaurants from the north part of Thailand between November and December 2021.

### Isolation and identification

Each sample from 0.1% peptone water was transferred to 5 mL of trypticase soy broth (TSB; HiMedia, Mumbai, Maharashtra, India) with 10% NaCl and 1% sodium pyruvate and incubated at 37 °C for 24 h. A loopful of this enrichment culture was inoculated onto Mannitol salt agar (MSA; HiMedia, Mumbai, Maharashtra, India) and incubated at 37 °C for 24 h. Cultures with yellow colonies were selected and sub-cultured on MSA to obtain pure colonies. Positive results from biochemical testing for catalase, DNase and coagulase identified *S. aureus* (*Joshi et al., 2017*). All *S. aureus* were confirmed by detecting the 16SrRNA gene using the methods outlined by *Kohner et al. (1999)*.

### MRSA screening and PCR confirmation

Methicillin-resistant *S. aureus* (MRSA) was identified by growing the strains on MSA containing 4 μg/ml of oxacillin, confirmed by disk diffusion using cefoxitin 30 μg and then the presence of the *mec*A gene was confirmed by PCR using specific primers (Table 1; *Kitti, Boonyonying & Sitthisak, 2011*). Genomic DNA was extracted using the boiling method and then used as the DNA template in PCR amplification. Amplification started with
**Table 1 Primers of PCR products for the *S. aureus* gene-specific.**

| Genes | Primer | Primer 5′->3′ | Reference |
|---|---|---|---|
| 16srRNA | 16sF | CGAAAGCCTGACGGAGCAA | *Kohner et al. (1999)* |
| | 16sR | AACCTTGCGGTCGTACTCCC | |
| *mecA* | mecAF | TGGCTATCGTGTCACAATCG | *Kitti, Boonyonying & Sitthisak (2011)* |
| | mecAR | CTGGAACTTGTTGAGCAGAG | |
| *icaAD* | icaADF | AATGTGCTTGGATGCAGATACTATC | *Seng et al. (2018)* |
| | icaADR | GAATCGTCATCTGCATTTGCA | |
| *cna* | CnaF | AAAGCGTTGCCTAGTGGAGA | *Arciola et al. (2005)* |
| | CnaR | AGTGCCTTCCCAAACCTTTT | |
| *fnbA* | FnaF | GATACAAACCCAGGTGGTGG | *Arciola et al. (2005)* |
| | FnaR | TGTGCTTGACCATGCTCTTC | |
| *bap* | Bap971F | CCCTATATCGAAGGTGTAGAATTGCAC | *Cucarella et al. (2001)* |
| | Bap971R | GCTGTTGAAGTTAATACTGTACCTGC | |
| *sea* | SEAF | GGTTATCAATGTGCGGGTGG | *Mehrotra, Wang & Johnson (2000)* |
| | SEAR | CGGCACTTTTTTCTCTTCGG | |
| *seb* | SEBF | GTATGGTGGTGTAACTGAGC | *Mehrotra, Wang & Johnson (2000)* |
| | SEBR | CCAAATAGTGACGAGTTAGG | |
| *sec* | SECF | AGATGAAGTAGTTGATGTGTATGG | *Mehrotra, Wang & Johnson (2000)* |
| | SECR | CACACTTTTAGAATCAACCG | |
| *sed* | SEDF | CCAATAATAGGAGAAAATAAAAG | *Mehrotra, Wang & Johnson (2000)* |
| | SEDR | ATTGGTATTTTTTTTCGTTC | |
| *see* | SEEF | AGGTTTTTTCACAGGTCATCC | *Mehrotra, Wang & Johnson (2000)* |
| | SEER | CTTTTTTTTCTTCGGTCAATC | |

Note:
Nucleotide sequences of PCR products for the *S. aureus* gene-specific oligonucleotide primers used in this study.

pre-denaturation at 94 °C for 5 min, followed by 35 cycles of amplification (denaturation at 92 °C for 20 s, annealing at 58.5 °C for 20 s, and extension at 72 °C for 30 s), and one cycle of final extension at 72 °C for 7 min. The PCR products were analyzed by 1% agarose gel electrophoresis.

## Biofilm formation

A Congo red agar (CRA) test was performed to determine biofilm production (*Freeman, Falkiner & Keane, 1989*; *Seng et al., 2017*). All *S. aureus* isolates were inoculated onto the CRA plates and incubated for 24–48 h at 35 °C. Biofilm production was identified by the color change of the colonies, with a change from red to black or very black indicating biofilm production. Colonies that remained red were identified as non-biofilm producers.

## Molecular detection of biofilm-related adhesion genes

Table 1 shows the primers used to detect the *S. aureus* biofilm-related genes: *bap*, *fnb*A, *ica*AD, and *cna*. DNA templates were extracted from each sample using the boiling method: the pellet of bacterial cell was resuspended in 40 μl of molecular-grade water, subjected to boiling at 95–100 °C in a water bath for 10 min, and centrifuged at 6,000 rpm

for 5 min. The supernatant was used as a DNA template for PCR reaction. The reaction was carried out with a total volume of 25 μl including a 2 μl DNA template, 20 pmol of each primer, and 12.5 μl of one PCR (GeneDireX, Inc., Taoyuan, Taiwan), with the total volume adjusted to 25 μl with sterile distilled water. Amplification began with a pre-denaturation step at 94 °C for 2 min, followed by 30 cycles of 94 °C for 20 s, annealing for 30 s at 58 °C for *ica*AD, at 55 °C for *fnb*A gene, at 52 °C for *cna* and at 50 °C for *bap* gene, then extension at 72 °C for 1 min and final extension at 72 °C for 5 min. The PCR products were analyzed by 1% agarose gel electrophoresis.

## Molecular detection of enterotoxin genes

In this study, we used five primer sets for five classical enterotoxin genes (Table 1) following the methods outlined by *Mehrotra, Wang & Johnson (2000)*, with minor modifications. The boiled cell lysates were used as the DNA template for PCR reactions. The reaction was carried out with a total volume of 25 μl including a 2 μl DNA template, 20 pmol of each primer, and 12.5 μl of one PCR (GeneDireX, Inc., Taoyuan, Taiwan), with the total volume adjusted to 25 μl with sterile distilled water. Amplification started with initial denaturation at 94 °C for 5 min, followed by 35 cycles of amplification (denaturation at 94 °C for 5 min, annealing at 50 °C for 2 min, and extension at 72 °C for 1 min), ending with a final extension at 72 °C for 7 min. Amplicons were analyzed by 1% agarose gel electrophoresis.

## Statistical analysis

Excel, the Statistical software for data science (STATA) and the Statistical Package for Social Science (SPSS) were used for the statistical analysis. The prevalence of *S. aureus* isolated from food handlers and other food contact surfaces was evaluated using an Odds Ratio (OR) with a 95% Confidence Interval (CI). The Fisher's exact test was performed by STATA and a $p$-value less than 0.05 was considered statistically significant. The association between biofilm-related *S. aureus* was investigated using the Phi-coefficient in SPSS and the pairs correlation between different genes. The sample size was 200, and the degree of freedom was 1. The correlation was considered statistically significant when the $p$-value was less than 0.05 ($p$-value < 0.05).

## RESULTS

### Prevalence of *S. aureus* on kitchen utensils and the hands of food handlers

Among 650 swab samples collected from food contact surfaces in restaurants, we identified 267 yellow colonies on MSA as presumptive *S. aureus* that we then confirmed with biochemical testing and PCR analysis. Two hundred isolates of *S. aureus* were detected. Among all swab samples, those collected from the hands of food handlers were the main source of *S. aureus* contamination, accounting for 78% of all *S. aureus* isolates found (117 from 150 samples from hands of food handlers) (OR 67.36, 95% CI [25.95–173.77]; Fisher's exact test, $p$ < 0.001; Fig. 1). The results showed that chopping boards, plates,

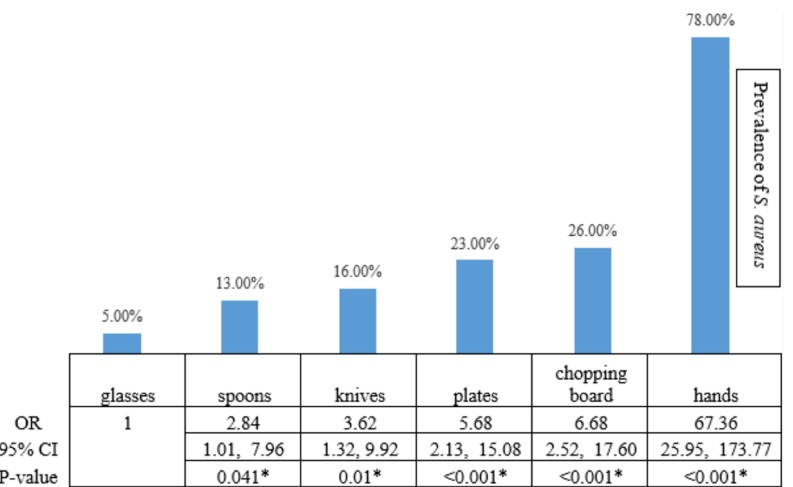

**Figure 1 The statistical analysis.** The statistical analysis demonstrated that the risk of *S. aureus* prevalence on hands, chopping boards, plates, knives, and spoons was higher than the risk on glasses, with a statistical significance level of 0.05. An asterisk (*) indicates statistical Statistical significance from the Fisher's exact text (*p*-value of <0.05 was considered as statistically significant).

knives, and spoons had a significantly higher prevalence of *S. aureus* contamination than glasses.

## Isolation of methicillin-resistant *S. aureus*

A total of 200 *S. aureus* isolates from food contact surfaces were analyzed for methicillin resistance by observing the growth of the isolates on MSA (containing oxacillin and cefoxitin), followed by *mecA* gene detection. All isolates were methicillin-sensitive strains, and the *mecA* gene was absent in all strains.

## Biofilm formation on CRA agar

Based on the CRA biofilm formation test, 49 isolates (24.5%) created biofilms as black colonies on Congo red agar, while 151 isolates (75.5%) were classified as non-biofilm producing *S. aureus*. Of the 49 biofilm-producing *S. aureus*, 36 were from isolates from the hands of food handlers (36/117; 30.77% of *S. aureus* isolates taken from the hands of food handlers). The lowest prevalence of biofilm-producing *S. aureus* was observed on the chopping boards (2/26; 7.69% of *S. aureus* isolates taken from chopping boards; Fig. 2). Therefore, we compared the risk of a biofilm-producing *S. aureus* on the hands of food handlers and other food contact surfaces compared with the risk from a chopping board. The statistical result showed that the risk of biofilm-producing *S. aureus* on the hands of food workers was significantly higher than on chopping boards (OR 5.33, 95% CI [1.20–23.78]; Fisher's exact test, *p*-value < 0.05). However, the risk of biofilm-producing *S. aureus* on spoons, glasses, knives, and plates was not significantly different from that on a chopping board (Fig. 2).
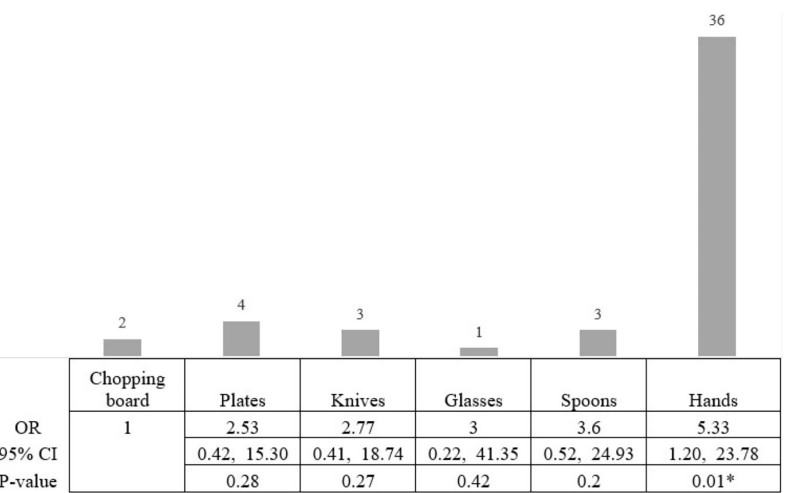

**Figure 2 The statistical analysis showed the risk of biofilm-producing *S. aureus*.** The statistical analysis showed that the risk of biofilm-producing *S. aureus* on hands was higher than on chopping boards, with a statistical significance level of 0.05. The highest risk of biofilm-producing *S. aureus* was on the hands (OR 5.33, 95% CI [1.20–23.78]; Fisher's exact test, *p*-value < 0.05). An asterisk (*) indicates statistical significance from the Fisher's exact text (*p*-value of <0.05 was considered as statistically significant).

**Table 2 Presence of biofilm formation and biofilm-related genes in *S. aureus*.**

| Biofilm | Sources | | | | | |
|---|---|---|---|---|---|---|
| | **Hands (n = 117) (%)** | **Chopping board (n = 26) (%)** | **Plate (n = 23) (%)** | **Knife (n = 16) (%)** | **Spoon (n = 13)** | **Glass (n = 5) (%)** |
| Red (%) | 81 (69.23) | 24 (92.31) | 19 (82.61) | 13 (81.25) | 10 (76.92) | 4 (80.00) |
| *ica*AD (%) | 0 | 0 | 0 | 0 | 0 | 0 |
| *fnb*A (%) | 5 (6.17) | 1 (4.17) | 3 (15.80) | 1 (7.69) | 1 (10.00) | 0 |
| *cna* (%) | 2 (2.47) | 0 | 0 | 0 | 0 | 0 |
| *bap* (%) | 0 | 0 | 1 (5.26) | 0 | 0 | 0 |
| co-presence *fnb*A+*cna* | 1 (1.24) | 1 (4.17) | 1 (5.26) | 0 | 0 | 0 |
| None | 73 (90.12) | 22 (91.66) | 14 (73.68) | 12 (92.31) | 9 (90.00) | 4 (100.00) |
| Black (%) | 36 (30.77) | 2 (7.69) | 4 (17.39) | 3 (18.75) | 3 (23.08) | 1 (20.00) |
| *ica*AD (%) | 9 (25.00) | 1 (50.00) | 1 (25.00) | 1 (33.33) | 0 | 0 |
| *fnb*A (%) | 1 (2.78) | 1 (50.00) | 0 | 0 | 0 | 0 |
| *cna* (%) | 3 (8.33) | 0 | 0 | 1 (33.33) | 0 | 0 |
| *bap* (%) | 0 | 0 | 0 | 0 | 0 | 0 |
| co-presence *ica*+*fnb*A+*cna* (%) | 3 (8.33) | 0 | 0 | 0 | 0 | 0 |
| co-presence *ica*+*fnb*A (%) | 7 (19.44) | 0 | 0 | 1 (33.33) | 2 (66.67) | 0 |
| co-presence *ica*+*cna* (%) | 1 (2.78) | 0 | 0 | 0 | 0 | 0 |
| None | 12 (33.34) | 0 | 3 (75.00) | 0 | 1 (33.33) | 1 (100.00) |

**Note:**
Red colonies on CRA plates were classified as non-biofilm producers, black colonies were identified as biofilm-producers.
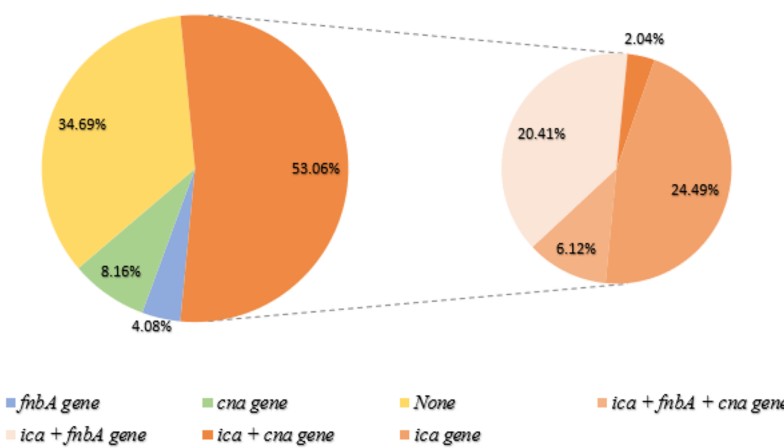

**Figure 3 Biofilm-producing *S. aureus* and biofilm-related gene.** Of the 49 biofilm-producing *S. aureus*, 36 isolates (53.06%) were detected to have the *ica* gene (orange). The right panel represents the co-presence of genes: (*ica + fnb*A + *cna* gene = 6.12%), (*ica + fnb*A = 20.41%), (*ica + cna* = 2.04%) and (only *ica* gene = 24.49%). None of the genes were found in biofilm producers (34.69%; Yellow). Green and blue represent genes *cna* and *fnb*A, respectively.   

## Genotypic profile of *S. aureus* isolates

The prevalence of biofilm-related genes was determined in all *S. aureus* isolates (Table 2); *ica*, *fnb*A, and *cna* genes were found in 26, 29, and 13 isolates, respectively, whereas the *bap* gene was detected in only one non-biofilm producing isolate. Out of the 49 biofilm-forming *S. aureus*, three adhesion genes were found in three isolates, with the *ica* gene identified in the most (Fig. 3). The prevalence of adhesion genes in non-biofilm producing *S. aureus* is shown in Fig. 4. Of the 151 non-biofilm producing *S. aureus*, biofilm-related genes were not detected in 134 isolates (88.74%). There was a strong positive and significant correlation between *ica* and *fnb*A (Phi. = 0.39, $p < 0.01$) and between *fnb*A and *cna* (Phi. = 0.24, $p < 0.01$; Table 3).

Two-hundred isolates of *S. aureus* were detected along with their classical enterotoxin genes (*sea*, *seb*, *sec*, *sed*, and *see*). The *sec* gene was found in four isolates (2%) and the *sed* gene was found in six isolates (3%). The *sec* gene was found on three chopping boards and one knife. The *sed* gene was found on three chopping boards, two hands and one knife.

## DISCUSSION

Two hundred isolates of *S. aureus* were found among 650 swab samples collected from restaurant food contact surfaces (hands, plates, spoons, knives, chopping boards, and glasses). Swab samples collected from the hands of food handlers were found to be the primary source of *S. aureus* isolates (78%). *S. aureus* is a symbiotic bacterium that has been found on human skin, in nostrils, and in the gastrointestinal tract. A study of nasal carriers working in food production environments found that the *Staphylococcus* carrier could contaminate food through contact with respiratory secretions, resulting in staphylococcal food poisoning (*Bencardino, Amagliani & Brandi, 2021*). *Ogidi, Oyetayo & Akinyele (2016)* reported the highest occurrence of *S. aureus* (23.30%) found on street foods sold in Akure Metropolis, Nigeria, amongst the other bacterial contaminants. This could be because of

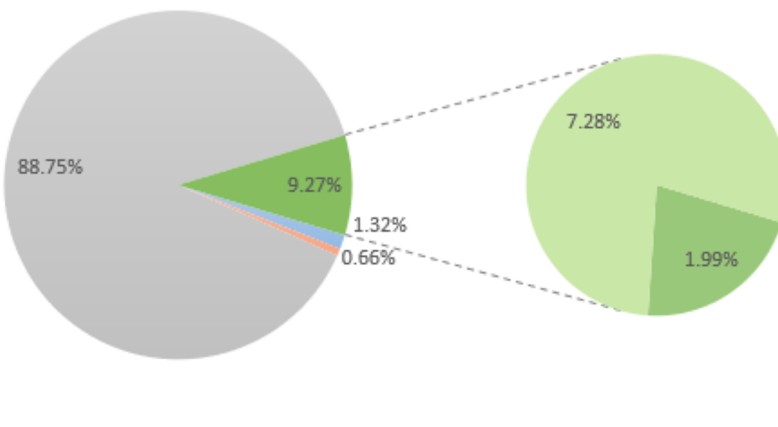

■ *cna gene* ■ *bap gene* ■ *None* ■ *fnbA + cna gene* ■ *fnbA gene*

**Figure 4 Non-biofilm producing *S. aureus* and biofilm-related gene.** Of the 151 non-biofilm pro-ducing *S. aureus*, 134 isolates (88.74%) lacked biofilm-related genes. Only one isolate (0.66%; pink) had the *bap* gene, and 1.32% of isolates had the *cna* gene (blue). Green (14) represents the *fnbA* gene and three of these samples also had *cna* gene (1.99%).

**Table 3 The association between biofilm-related genes in this study.**

|  |  | *ica* | *fnbA* | *cna* | *bap* |
|---|---|---|---|---|---|
| *ica* | Phi-coefficient | 1 |  |  |  |
|  | Sig. (2-tailed) | – |  |  |  |
| *fnbA* | Phi-coefficient | 0.39** | 1 |  |  |
|  | Sig. (2-tailed) | 0.00 | – |  |  |
| *cna* | Phi-coefficient | 0.14* | 0.24** | 1 |  |
|  | Sig. (2-tailed) | 0.05 | 0.00 | – |  |
| *bap* | Phi-coefficient | −0.03 | −0.03 | −0.02 | 1 |
|  | Sig. (2-tailed) | 0.70 | 0.68 | 0.79 | – |

**Notes:**
** A *p*-value less than 0.01 ($p < 0.01$) was considered as statistically.
* Correlation is significant at the 0.05 level (2-tailed), Sig., Significance.
Using a Phi-coefficient, there was strong positive significant correlations between *ica* and *fnbA* (Phi. = 0.39, $p < 0.01$) and between *fnbA* and *cna* (Phi. = 0.24, $p < 0.01$).

unwashed hands or other poor hygiene practices of the food vendors. According to the requirements of the Microbiological Quality Criteria for Food and Food Contact Containers issued by the Department of Medical Science, Ministry of Public Health, Thailand (2017), *S. aureus* should not be found on any containers contacting foods or on food handlers. In this experiment, hands were the most contaminated with *S. aureus*, followed by chopping boards, most of which are made of wood, emphasizing the risks associated with cross-contamination. Forty-nine biofilm-forming *S. aureus* (49/200) were identified, accounting for 24.5% of the total *S. aureus* strains found, most of which (36/49; 73.47%) were isolated from the hands of food handlers. *Kim et al. (2017)* reported that the formation of biofilm affected the attachment of *S. aureus* to food contact surfaces. The formation of biofilms allowed bacteria to adhere to food contact utensils, creating barriers to cleaning (including disinfection), which has become a problem in the food

industry (*Avila-Novoa et al., 2018*). Because of their low antibiotic sensitivity, staphylococci in clinical isolation are difficult to treat clinically (*Cramton, Gerke & Götz, 2001*). All isolates identified in this study were methicillin-susceptible strains. However, resistant strains are detected in hospitals and community settings, indicating *S. aureus* may pose a greater threat in the future (*Schito, 2006*; *Phokhaphan et al., 2017*; *Hamdan et al., 2022*).

To determine biofilm production, we used both phenotypic (CRA) and genotypic (PCR) analyses. The *ica* gene and phenotype were both detected in 26 *S. aureus* isolates. The remaining 23 samples were phenotypically positive but lacked the *ica* locus gene, implying that *S. aureus* may use which another operon to produce biofilm (*Marques et al., 2017*). We discovered no relationships between the other genes and biofilm formation (*fnb*A, *cna*, and *bap*). The results of this experiment are consistent with those of previous studies, indicating that the presence of the *cna*, *fnb*A, and *bap* genes is not consistent with biofilm formation (*Płoneczka-Janeczko et al., 2014*; *Tangchaisuriya et al., 2014*; *Seng et al., 2017*).

The enterotoxin genes detected in this study were *sec* and *sed*. *S. aureus* strains with *sec* were identified on a chopping board and knife that usually have direct contact with meat. Previous reports showed that *sec* was most common in animals (*Pinchuk, Beswick & Reyes, 2010*), implying that *S. aureus* carrying the *sec* gene in our study might come from meat contamination, although other sources including the washing water might be sources of contamination (*Lin et al., 2016*). Enterotoxin D is the second most common cause of food poisoning in the world (*Pinchuk, Beswick & Reyes, 2010*) and also found in MRSA strains (*Schmitz et al., 1997*). In 2007, researchers report 3 years of findings of the presence of *S. aureus* in meat and dairy products, and SED was detected in most strains (*Normanno et al., 2007*). In 2020, the classical enterotoxin genes (*sea*, *seb*, *sec* and *sed*) were detected from MRSA isolated from food handlers (*Ahmed, 2020*). One of the most important SE properties is heat stability. A small amount of this toxic substance can cause disease (*Pinchuk, Beswick & Reyes, 2010*). Most genes coded for staphylococcal enterotoxins are located on mobile genetic elements (*Pinchuk, Beswick & Reyes, 2010*), which could be easily transferred to other related strains *via* horizontal gene transfer events. Therefore, further research on enterotoxin expression on food contact surfaces is needed as well as the development of molecular techniques to detect virulence genes to prevent foodborne illness outbreaks.

This is an observational study and hence there was no attempt to standardize or influence cleansing and hygiene practices prior to taking samples. This limits the reproducibility of the study but reflects these "real world" situation. The study was performed in a single period of 2 months and is not necessarily reflecting the situation at other, perhaps hotter times of the year.

## CONCLUSIONS

Food contamination is one of the major concerns for consumer health safety especially in local restaurants where food regulations have not properly implied compare to standard restaurant in the main cities of Thailand. For this study, 650 swabs testing were conducted

on surfaces of kitchen utensils as well as on hands of food handlers working in local restaurants in the north part of Thailand. The swap were conducted on dry surfaces after normal cleansing process of each restaurants and individual food handlers, none specific cleaning method provided.

Two hundred *S. aureus* were identified from 650 samples, where the highest were detected on the hands of food handlers (78%). *S. aureus* has the ability to form bilfilms, which enable the distribution of *S. aureus* on foods *via* cross-contamination events. However, not all of *S. aureus* can form bilofilm. In this study, using the CRA method, 49 out of 200 *S. aureus* were identified as biofilm-producing strains, of which most were *ica*AD gene (26 out of 49). In addition, MRSA screening and PCR confirmation were also conducted; however, no MRSA has been found. Lastly, additional experiments were conducted to see if any existing enterotoxin genes existed, and two classical enterotoxin genes, *sec* and *sed*, were found in four and six of the *S. aureus* isolates, respectively, from hands and utensils.

*S. aureus* is one of the most common pathogens responsible for food poisoning. The discovery of *S. aureus* enterotoxin genes and biofilm-producing strains on food contact surfaces indicates that these may be pathogen transmission sources. Therefore, more awareness of food regulation and education in food hygiene should be carried out for all food handlers. A thorough washing of hands and other food-contact surfaces can reduce the risk of cross-contamination of food. The development of surveillance systems for staphylococcal food contamination would increase food safety. If any additional research is conducted, it may concentrate on the water quality in the area and the necessity of using the right hand soap or detergent in cleaning process to improve hygiene in local area where more hygiene regulations will be required for certified restaurants.

### Funding
This work was funded by a grant from the National Science, Research and Innovation Fund (NRSF) (2021: R2564B028). The funders had no role in study design, data collection and analysis, decision to publish, or preparation of the manuscript.

### Grant Disclosures
The following grant information was disclosed by the authors:
National Science, Research and Innovation Fund (NRSF): 2021: R2564B028.

### Competing Interests
Sutthirat Sitthisak is an Academic Editor for PeerJ.

### Author Contributions
- Kannipa Tasanapak conceived and designed the experiments, performed the experiments, analyzed the data, prepared figures and/or tables, authored or reviewed drafts of the article, and approved the final draft.

- Siriwat Kucharoenphaibul performed the experiments, authored or reviewed drafts of the article, and approved the final draft.
- Jintana Wongwigkarn performed the experiments, prepared figures and/or tables, and approved the final draft.
- Sutthirat Sitthisak conceived and designed the experiments, authored or reviewed drafts of the article, and approved the final draft.
- Rapee Thummeepak analyzed the data, prepared figures and/or tables, and approved the final draft.
- Plykaeow Chaibenjawong conceived and designed the experiments, prepared figures and/or tables, and approved the final draft.
- Wassana Chatdumrong performed the experiments, authored or reviewed drafts of the article, and approved the final draft.
- Kunsuda Nimanussornkul analyzed the data, prepared figures and/or tables, and approved the final draft.

### Human Ethics

The following information was supplied relating to ethical approvals (*i.e.*, approving body and any reference numbers):

This study was approved by the Naresuan University Institutional Review Board (NU-IRB).

### Data Availability

The raw data, all the sources of *S. aureus* contamination, are available in the Supplemental File.

### Supplemental Information

Supplemental information for this article can be found online at http://dx.doi.org/10.7717/peerj.15824#supplemental-information.

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
