# Peer review of "Prevalence and virulence genes of Staphylococcus aureus from food contact surfaces in Thai restaurants"

_PeerJ, doi:10.7717/peerj.15824_

## Round 0.1 · original submission · Major Revisions

Please authors, kindly consider all the comments raised by the scholarly reviewers.
Please pay particular attention to the comments from Reviewer 3, and ensure to diligently address them.

Major revision is needed, given the cumulative breadth and depth of comments.
Look forward to seeing your revised manuscript.

·

Basic reporting

In the Introduction, line 47 there is an extra semicolon and please provide more information on Biofilm from lines 54-59.

Experimental design

1. As far as I understand through this article there were no reference strains mentioned to conform to the experiment mentioned in line 77 and for the slime-producing strains and non-slime-producing strains, Kindly add the necessary information.


2. In the CRA test have you received only red or black colonies? or did you have any intermediate-coloured colonies?
Please explain the DNA extraction step in line 90

3. Maybe the author can rephrase line 101 as it lacks a connection with the procedure.

Validity of the findings

Given the statistical results for the CRA biofilm formation test and PCR, Please include the figure for the plates and the gel pictures respectively. Given below are the details for the gel pictures.

As per lines 120 and 121 can you please include the gel picture for the PCR to identify the ica, fnbA, can genes and bap gene?

Please elaborate on lines 127-128 and provide the reference figure.

Additional comments

This is an interesting study about Staphylococcus aureus and biofilm formation. The paper is generally well-written and structured. However, in my opinion, the paper has some shortcomings in regard to data analyses and text. I have provided a few remarks on the methodology and the data interpretation and I would recommend adding the data for the PCR and other tests in the Results section for the paper to be published.

Reviewer 2 ·

Basic reporting

The manuscript needs an English language editing.
The references need to be updated.

Experimental design

The hypothesis is not well defined.
The methdology lack of some necessary information

Validity of the findings

The results needs to be improved and statistical analysis should be revised.
The conclusion should be revised according to the results.

Additional comments

All my comments are presented in the attached annotated file that the authors should take into consideration.

Annotated reviews are not available for download in order to protect the identity of reviewers who chose to remain anonymous.

Reviewer 3 ·

Basic reporting

1. The English language needs improvement, especially in grammar, professional wording, and connection between sentences and paragraphs. I recommend having a colleague fluent in English and familiar with the content proofread and edit the manuscript. 
2. Although the authors presented the results, they did not include rationales for analysis or interpretations of the results. For example, what was the authors' rationale for testing the association between biofilm formation genes? What did Phi-coefficients between some of the biofilm formation genes indicate? Moreover, what about the lack of association between bap and other biofilm genes? Again in lines 127-128, was it surprising to find 2 SEs in 200 samples? How did the results compare to prior studies (sampling or detection methods, geographic locations, surfaces)? Do sec and sed genes differ from other enterotoxin genes regarding pathogenicity or prevalence? 
3. The authors did not present the results in figures. One way to improve data visualization is to include pie charts to show the breakdown of S. aureus from different sources, the percentage of each biofilm formation, or enterotoxin genes. The authors should also include representative images of agar plates. 
4. The introduction need an expansion on the knowledge gaps.
5. Although I agree that S. aureus can cause disease when ingested, infections do not always occur because of immunity. The authors need to mention it in the introduction, possibly from the angle of the susceptible population.
6. The discussion should focus on the experimental results. Including a new result that one cannot rely on the presence of biofilm formation genes to infer a phenotype that was not mentioned earlier is very confusing. I suggest the author include analysis in the results to give the reader an idea of how to interpret the PCR results. 
7. The statistical analysis in lines 121-123 should be removed from the main text and included in the methods section. More details are also necessary to help the readers understand the analyses.

Experimental design

1. The authors only examined the presence of enterotoxin genes, but it is unclear whether they were expressed or functional. More experiments are necessary.

Validity of the findings

1. In lines 177-179, the authors overstated the association between the presence of biofilm formation or enterotoxin genes and increased cases of food poisoning. The authors did not provide experimental or literature evidence on enterotoxin production. Additionally, the presence of biofilm formation genes is irrelevant to the phenotype (lines 157-160).

Reviewer 4 ·

Basic reporting

Most of the references are too old

Experimental design

The methods were not well detailed

Validity of the findings

The conclusion is too general and not directed to the study presented

Additional comments

The study is important in Public Health because Staphylococcus aureus is the leading cause of human health
Most of the references in the work are old and should be replaced with more recent ones. There is an avalanche of works on S. aureus virulence factor expression and genotype
The abstract did not mention the food contact surfaces and the study area
Line 45: check this statement ‘annual basis’ the reference cited is old
Ines 45-47: virulence factors such as extracellular factors, toxins, adhesion, biofilm formation, and resistance to phagocytosis – extracellular factors, adhesion biofilm formation and resistance to phagocytosis are not virulence factors but rather virulence factors. Correct the sentence
line 47: remove “;” and close the space
line 47-48: remove refs older than 2015
Line 50-51: is it that the heating process may not be sufficient to destroy the toxins or that the toxins are heat-stable?
Line 51: food handles or food handlers?
Line 55: Remove the reference (Hamadi et al., 204; it is not complete and old
Line 60: is it food handling equipment or food processing equipment or food contact article (as in line 138)?
The justification for the study is not well-articulated. For example, what studies have been conducted on S. aureus in the study area?
Line 74: The area to be swabbed was chosen – what is the dimension of the swabbed surfaces
Line 75-76: Remove “After taking the swab and putting it back in the buffer, the microbiologist will examine it further in the laboratory”
Line 73-78: The isolation protocol was not well described, for example, how were the isolates purified and the incubation conditions.
Phenotypic biofilm assay should come before the molecular characterization
Line 78: Molecular identification should be a subheading and the protocol described briefly under it
The result should have two subheadings – occurrence of S. aureus in kitchen utensils and Genotypic profile of S. aureus isolates
Line 121-122: Statistical analysis should be a subheading and the statistical tests and variables tested described under it
127-128: 5, 2 and 4 should be written in words
Line 137: Staphylococcus should begin with a capital letter
Line 132-144: The first line is a repetition of the result. The remaining sentences are not well related to the findings
Line 146: Staphylococci should begin with a small letter
Line 164-165: check the grammar
Generally, the discussion is not well articulated, and there are many grammatical errors that need to be improved.
Conclusion: The conclusion is too general and not directed to the study presented.

---

## Round 0.2 · Minor Revisions

Kindly address the minor corrections observed by the reviewers. Thank you very much

·

Basic reporting

No Comments

Experimental design

No comments

Validity of the findings

No comments

Additional comments

The authors have revised the manuscript as per the recommendations and they have included the necessary details.

Reviewer 2 ·

Basic reporting

The manuscript is etter like this

Experimental design

the research question is well defined and the methods are well described

Validity of the findings

The conclusion needs to be revised. you need to add the most imortant results of your study

Additional comments

I have other "minor" comments:
1. Revise the reference of line 140
2. P value should be written as p (lower case and itaic) in all the manuscript
3. Line 219: biofilm
4. Line 220: replace "caused" by "allowed"
5. Line 223-226: you should explain the sensitivity not the resistance since all your strains were sensitie (there is a contradiction in your explanaion)
6. Line 239: add a reference for the sentence. The same for the following
6. You should add te limitaions of the study
7. the cnclusion should be revised: "Underkooked...sources". This seemed to be a discussion not a conclusion. delete please
You should reduce the number of figures: delete figures 2 and 6

Reviewer 4 ·

Basic reporting

I suggest the title of the manuscript be changed to: “Prevalence and virulence genes of Staphylococcus aureus from food contact surfaces in Thai restaurants” or “Prevalence and virulence genes of Staphylococcus aureus from food contact surfaces in restaurants in Thailand”
33: antibiotic resistant – antibiotic-resistant
34: untreatable infection – difficult-to-treat infection
36: for hormone – confusing
51-52: and has been detected in 52 subclinical mastitis in dromedary camels in Algeria (Hadef, Aggad & Hamad, 2018) – this statement is not appropriate at this point. Why camels whereas the work is focused on kitchen utensils
76: Staphylococci – start with small letter
77: of these strains were – which strains? If it is the MR-CONS , mention it

Experimental design

93-93: “all serving utensils and food handler hands were cleaned using the methods commonly used in their restaurants” – briefly describe the cleansing methods to ensure reproducibility/preproducibility of the study. It is also important to state whether the surfaces were allowed to dry or not before swabbing was done. The presence of S. aureus on the surfaces before use may indicate ineffective cleansing methods or that the water sources/handlers constituted the source of the organisms on the surfaces
104: Each sample was transferred to 5 mL of trypticase soy broth – this is not clear. Was the peptone water incubated and then the broth culture sub-cultured onto salt agar or was it unincubated and the swabs removed from it and inoculated into salt agar?
105: what is the basis of adding sodium pyruvate – is it the standard method
113: 30 ug of cefoxitin or 30ug/mL – 30 ug and 30ug/mL implies that different methods (diac diffusion and MIC, respectively) were employed
121: Reference for Congo red agar test
163: confirmation - check grammar

Validity of the findings

223-226: All isolates identified in this study were methicillin-susceptible strains, demonstrating S. aureusís remarkable ability to adapt to new types of antimicrobial agents. This resistance is moving from the hospital to the community, posing an even greater threat (Schito, 2006) – This is confusing because susceptibility was exhibited whereas resistance was being discussed.
227: Remove (already in the methodology)
230: Which another operon
Line 236-238: It is not be convincing enough that the sec-positive strains on the knife and chopping boards originated from meat since there was no statement showing how the equipment were washed prior to swabbing. It is very likely that same water was used in washing all the utensils.
239: Appropriate reference should back the statement on enterotoxin D
There is need for comparison of the findings with that of previous reports.
250-251: Uncooked meat and unwashed hands could be the main sources of contamination – This is better suited for discussion
254: Washing hands and other food contact surfaces articles well – Rephrase as “Thorough washing of….

---

## Round 0.3 · accepted · Accept

Thank you authors for your diligent revision of their work. After very careful check, I am very satisfied with the current revised manuscript and hereby approve it for publication. Thank you authors for finding PeerJ Life and Environment as your journal of choice. Look forward to your future scholarly contributions.
Congratulations